# Decreased incidence, virus transmission capacity, and severity of COVID-19 at altitude on the American continent

**Christian Arias-Reyes**[1], **Favio Carvajal-Rodriguez**[1], **Liliana Poma-Machicao**[1], **Fernanda Aliaga-Raduán**[1], **Danuzia A. Marques**[1], **Natalia Zubieta-DeUrioste**[2], **Roberto Alfonso Accinelli**[3], **Edith M. Schneider-Gasser**[4], **Gustavo Zubieta-Calleja**[2], **Mathias Dutschmann**[5], **Jorge Soliz**[1,2]*

**1** Faculty of Medicine, Université Laval, Institute Universitaire de Cardiologie et de Pneumologie de Québec (IUCPQ), Québec, Quebec City, Canada, **2** High Altitude Pulmonary and Pathology Institute (HAPPI-IPPA), La Paz, Bolivia, **3** Instituto de Investigaciones de la Altura, Universidad Peruana Cayetano Heredia, Lima, Peru, **4** Institute of Veterinary Physiology, Vetsuisse-Faculty University of Zurich, Zurich, Switzerland, **5** Florey Institute of Neuroscience and Mental Health, University of Melbourne, Melbourne, Victoria, Australia

* jorge.soliz@crchuq.ulaval.ca

**Data Availability Statement:** Raw, normalized, and adjusted data of COVID-19 cases are available at Figshare: https://doi.org/10.6084/m9.figshare.

## Abstract

The coronavirus disease 2019 (COVID-19) outbreak in North, Central, and South America has become the epicenter of the current pandemic. We have suggested previously that the infection rate of this virus might be lower in people living at high altitude (over 2,500 m) compared to that in the lowlands. Based on data from official sources, we performed a new epidemiological analysis of the development of the pandemic in 23 countries on the American continent as of May 23, 2020. Our results confirm our previous finding, further showing that the incidence of COVID-19 on the American continent decreases significantly starting at 1,000 m above sea level (masl). Moreover, epidemiological modeling indicates that the virus transmission rate is lower in the highlands (>1,000 masl) than in the lowlands (<1,000 masl). Finally, evaluating the differences in the recovery percentage of patients, the death-to-case ratio, and the theoretical fraction of undiagnosed cases, we found that the severity of COVID-19 is also decreased above 1,000 m. We conclude that the impact of the COVID-19 decreases significantly with altitude.

## 1. Introduction

On March 11, 2020, the coronavirus disease 2019 (COVID-19) was declared a pandemic by the World Health Organization [1, 2]. In late April, the health crisis began to ease in Asia and Europe [3–5], whereas case numbers began to rise in American countries. The first cases of COVID-19 on the American continent were reported in Canada on January 15[th] and in the United States on January 20[th]. Before February 25[th], Brazil was the first affected country in Latin America. From that date until July 7[th], the Pan-American Health Organization (PAHO) [6] reported 6,004,685 confirmed cases and 268,828 deaths from COVID-19 in the American continent [6]. The spread of the virus was so fast in the region, that it has now become the

12685478. Raw epidemiological daily data of COVID-19 of Argentina, Bolivia, Colombia, Ecuador, and Peru are available at Figshare: https://doi.org/10.6084/m9.figshare.12685523.

**Funding:** The author(s) received no specific funding for this work.

**Competing interests:** The authors have declared that no competing interests exist.

epicenter of the disease, with United States (1st), Brazil (2nd), Peru (5th), Chile (6th), Mexico (9th), Colombia (19th) and Canada (20th) among the top twenty countries with the highest number of confirmed cases in the world on July 7th, 2020 [7].

Although American countries had more time and information to prepare for the pandemic than did Europe and Asia, with few exceptions, weaker public health systems, late political responses, and complex cultural and social conditions (i.e., poorly respected quarantines in most countries) have led to a major public health crisis in the continent. A crucial aspect for the spread of the virus is overcrowding in large cities. Clear examples of this fact are the cities of New York (8.1 million people—10,194 people/$km^2$—214,371 cases) [8], Montreal (1,8 million people—4,517 people/$km^2$—27,438 cases) [9], and Rio de Janeiro (6, 7 million people—5,597 people/$km^2$—33,695 cases) [10] (data as of July 7). Furthermore, this scenario is particularly critical in the poorer districts of those cities, where large numbers of people occupy the same housing units [11]. Remarkably, however, other densely populated metropolises with large slum areas, but located above 1,000 meters above sea level (masl), such as México city (8 million people—26,000 people/$km^2$—53,423 cases) [12], Bogotá (7,4 million people—4,310 people/$km^2$—36,554 cases) [13] and La Paz (2,3 million people—2,676 people/$km^2$—4,413 cases) [14] (data as of July 7), seem to show lower incidences of COVID-19. This observation is of crucial importance since in American countries, more than 120 million people live at an altitude higher than 1,500 masl (defined as moderate altitude—MA), and more than 35 million people live at an altitude higher than 2,500 masl (defined as high altitude—HA) [15]. Furthermore, the capitals several American countries i.e., Bolivia, Brazil, Colombia, Costa Rica, Ecuador, Guatemala, and Mexico are located above 1,000 masl (S1 Table).

The impact of altitude on a potentially decreased virulence of SARS-CoV-2 was previously reported by our team. In fact, the global data analysis, which included detailed information from the Tibetan Autonomous Region of China, Bolivia, and Ecuador, suggested that the infection rates for the SARS-CoV-2 virus decrease significantly above 2,500 masl [16]. Subsequent reports from other research groups supported this observation [17–19]. However, being aware that the course of the pandemic changes from day to day and that more detailed statistical analyses are required, in this new study, we analyzed the epidemiological data from 23 countries in the American continent as of the 23rd of May 2020. Our results show that the incidence of COVID-19, the virus transmission rate, and the severity of COVID-19 decrease significantly above 1,000 masl.

## 2. Methods

### 2.1. Data sources

Supplementary S2 Table shows the list of data sources used to collect information from COVID-19 cases through May 23rd, 2020. Data were collected at the finest administrative level possible according to these categories: country (1st level), state, province, or departamento (2nd level), city or county (3rd level). For Bolivia, Brazil, Canada, Colombia, French Guyana, Panama, and USA, we used information of the number of cases per city/county. For Argentina, Belize, Chile, Costa Rica, Cuba, Ecuador, El Salvador, Haiti, Honduras, Mexico, Paraguay, Peru, Puerto Rico, Dominican Republic, Uruguay, and Venezuela, we used information of the number of cases per state/province/departamento.

All locations with reported positive COVID-19 cases were associated with their respective geographic coordinates (latitude and longitude) using the OpenCage Geocoding API [20]. The altitude information for each location was extracted from the WORLDCLIM digital elevation model [21], and the population density data were assigned from the dataset of the CIESIN [22]. In cases where the value of population density was zero, the information was retrieved

from the national statistical institute of the corresponding country. The full list of these locations can be found at: https://doi.org/10.6084/m9.figshare.12685610.v1.

## 2.2. Assessment of incidence versus altitude

The number of COVID-19 cases by location (per city/county or per state/province/departamento) was normalized by population density (inhabitants per square kilometer), in accordance with previous studies that demonstrated a positive correlation between population density and the number of COVID-19 cases [23–26]. These data were then grouped by intervals of 100 meters of altitude. Finally, the natural logarithm (ln) of each normalized value was calculated. These data (referred as the number COVID-19 cases in the text) were used for all the analyses unless stated otherwise. The correlation between the number of COVID-19 cases per altitude interval and the altitude was analyzed using a Pearson correlation analysis (n = 51). Identical correlation analyses were performed for each of the 23 countries that were studied ($n_{Argentina}$ = 12; $n_{Belize}$ = 3; $n_{Bolivia}$ = 27; $n_{Brazil}$ = 17; $n_{Canada}$ = 13; $n_{Chile}$ = 14; $n_{Colombia}$ = 36; $n_{Costa\ Rica}$ = 20; $n_{Cuba}$ = 4; $n_{Dominican\ Republic}$ = 9; $n_{Ecuador}$ = 19; $n_{El\ Salvador}$ = 8; $n_{French\ Guyana}$ = 1; $n_{Haiti}$ = 8; $n_{Honduras}$ = 12; $n_{Mexico}$ = 15; $n_{Panama}$ = 13; $n_{Paraguay}$ = 5; $n_{Peru}$ = 17; $n_{Puerto\ Rico}$ = 8; $n_{Uruguay}$ = 2; $n_{USA}$ = 34; $n_{Venezuela}$ = 12).

The difference in COVID-19 incidence below and above 1,000 masl at continental level was calculated using a bidirectional random block ANOVA. The advantage of this type of statistical analysis is that considers the internal variability within each country in the incidence analysis. The dependent variable was the number of COVID-19 cases (at 2nd or 3rd administrative level and not grouped by altitude intervals); the grouping variable was the altitude (> 1,000 masl or < 1,000 masl), and the blocks were the 23 countries ($n_{>1,000\ masl}$ = 827; $n_{<1,000\ masl}$ = 3,659).

## 2.3. Evaluation of the virus transmission rate

The evaluation of the SARS-CoV-2 virus transmission rate was performed only for Argentina, Bolivia, Colombia, Ecuador, and Peru, as these countries applied similar strong early quarantines and provided daily epidemiological data at state/province/departamento level.

The COVID-19 daily data retrieved by state/province/departamento in each country since the first reported case until May 23 was classified into two groups: highlands (>1,000 masl) and lowlands (<1,000 masl). A deterministic SEIR model (Susceptible—Exposed—Infectious—Removed) was used to calculate the estimated number of susceptible, exposed, infected and removed (recovered + deceased) individuals [27]. The equations and parameters used in the model are described in S1 Appendix. The number of susceptible individuals was calculated as the total population minus the number of infected and exposed subjects. The initial number of infected was set to 1, and the number of exposed subjects was arbitrarily set to 30 (according to [28]). The contact rate (β), recovery rate (Ƴ), and the rate at which exposed individuals become infected (Ɛ) were calculated as conducted elsewhere [28]. The contact rate was estimated as the product of the "interaction frequency" and the "probability of transmission of the disease" (also referred in the text as transmission rate) [28]. We set the values of interaction frequency = 8.1 [28], infectious period = 7.5 days, and incubation period = 6 days [29]. Asymptomatic individuals were considered as non-infectious. Recovered individuals were considered to be immune to reinfection. The size of the population was considered unchanged during the modelled time lapse. For the highlands (>1,000 m) and lowlands (<1,000 m) of each country, the most probable value of probability of transmission was estimated by fitting the SEIR model data to the observed data using the maximum likelihood method. The (one standard deviation) confidence intervals of the estimated probabilities of transmission were calculated by the graphical Monte Carlo method as described somewhere else [30]. Finally, the basic

reproduction number ($R_0$) was calculated as the product of "*contact rate*" x "*probability of transmission*" x "*infection period*".

## 2.4. Assessment of COVID-19 severity

The severity of the disease was estimated based on the percentage of recovered patients and the death-to-case ratio. Indeed, a lower mortality rate per case and a higher percentage of recovered patients suggest a lower severity of the disease. Accordingly, the severity of the disease was evaluated for Argentina, Bolivia, Colombia, Ecuador, and Peru, as these countries applied similar containment measures and provided daily epidemiological data at the state/province/departamento level.

The death-to-case ratio and the percentage of recovered patients ([recovered patients/reported cases] * 100) for each country (except Ecuador) were calculated using the data from the last 10 days (from May 13th to 23rd) for the populations above and below 1,000 masl in two separate pools. The number of deaths and recoveries used to calculate these parameters were the summary of the values reported for all the populations above and below 1,000 masl correspondingly. The underlying rationale is that these parameters stabilize as the pandemic progresses. As this information was not available for Ecuador on the mentioned dates, these values were calculated only from the data of April 29 (the latest data available for this country).

The Wilcoxon signed-rank test was used to determine if the percentage of recovered patients ($n_{pairs} = 5$) and the death-to-case ratio ($n_{pairs} = 5$) were different between the highlands and the lowlands.

## 2.5. Assessment of undiagnosed cases

COVID-19 patients can be asymptomatic or symptomatic. According to the severity of the disease, symptomatic patients are classified as follows: 1) Mild: patients with very mild symptoms and without evidence of viral pneumonia or hypoxia; 2) Moderate: patients with signs of pneumonia (fever, cough, dyspnea, rapid breathing but with regular arterial $O_2$ saturation values); 3) Severe: patients with established lung damage, symptoms of pneumonia, cough, fever, dyspnea and hypoxia; and 4) Critical: patients with systemic extrapulmonary inflammation and the presence of large amounts of proinflammatory cytokines [31]. Since health policies in most countries in the American continent restricted the access to COVID-19 tests to people showing clear symptoms of infection or having a history of contact with infected people [32–34], in this study, we have made the reasonable assumption that the cases observed and reported officially include mainly symptomatic (mild + moderate + severe + critical). Of note, only a fraction of the mild and moderate cases is diagnosed. We performed a theoretical calculation to determine the number of undiagnosed cases (asymptomatic + undiagnosed symptomatic). To do so, we used the infection mortality rates (IFR) of New York ($IFR_{NY}$ = 1.4—considered to the date to be the most reliable calculated value available for this parameter [35]) to calculate the "Estimated Total cases" (= observed deaths*$IFR_{NY}$) [36] and the "Estimated Fraction of Undiagnosed cases" (= [calculated total cases—observed cases]/calculated total cases) [35]; Verity, Okell [37]. Data on the observed deaths and recovered patients were obtained from each country's official government website on May 23rd, 2020.

## 2.6. Ethics statement

All data used in this study were obtained from public sources, generally from the official government's websites or repositories of each country. Data was fully anonymized before we had access to it.

## 3. Results

### 3.1. The incidence of COVID-19 decreases above 1,000 masl in the American countries

The correlation between the number of COVID-19 cases and the altitude at continental level (data from 23 countries pooled together) revealed a strong negative correlation ($p < 0.0001$; r = -0.777) between these variables, underlining a decrease in the incidence of COVID-19 cases with increasing altitude (see Fig 1a). Furthermore, we remarked that a significant decrease in the number of COVID-19 cases (not grouped by intervals of 100 meters of altitude) started

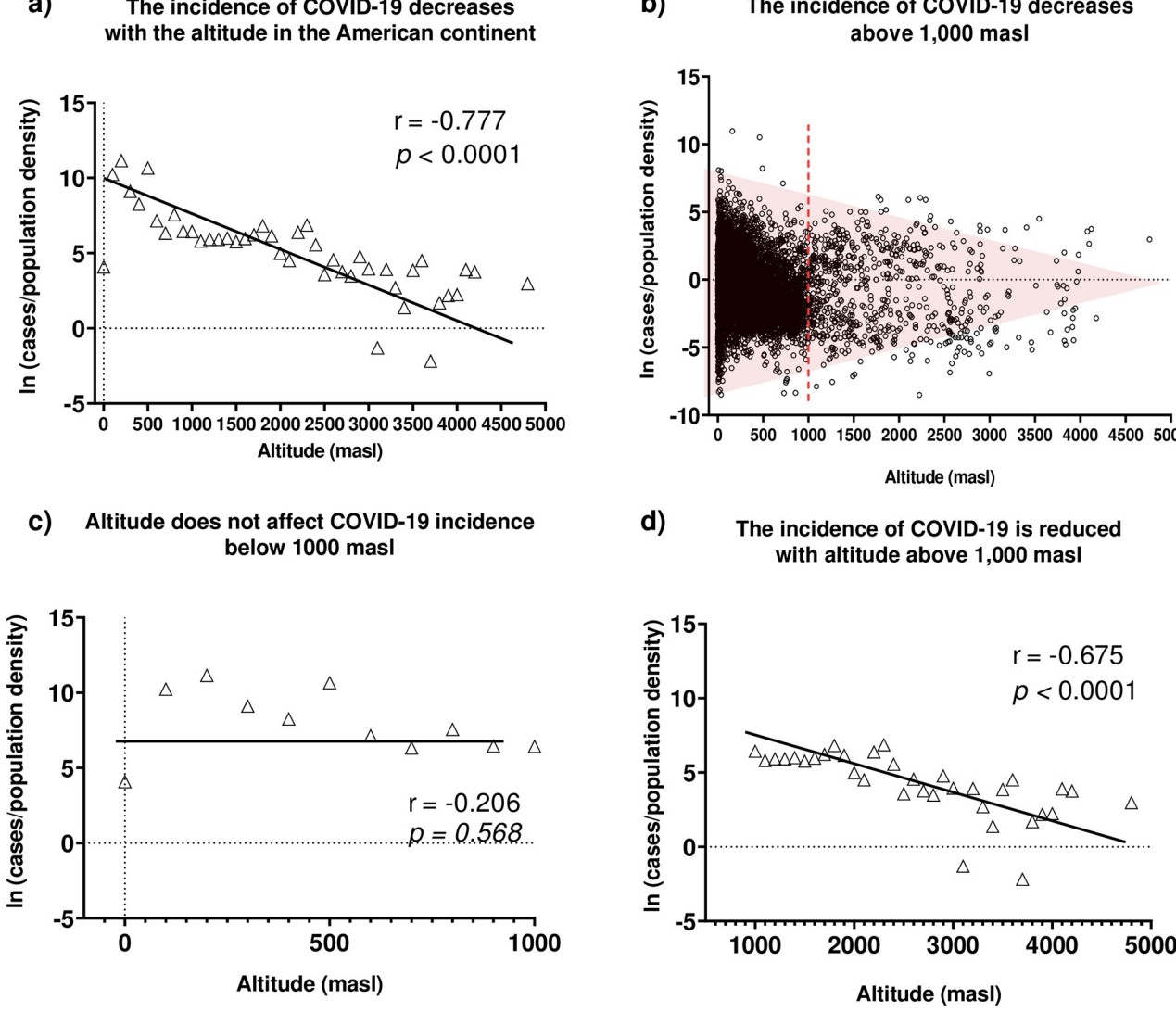

**Fig 1. The effect of altitude on the incidence of COVID-19 in the American continent.** Epidemiological data were retrieved on May 23. Data on population density were extracted from the dataset created by CIESIN [22] or the corresponding country's national statistics institute on May 23. Data were normalized by the population density of the same location and summed in intervals of 100 m of elevation. Raw, normalized, and adjusted data are available at https://doi.org/10.6084/m9.figshare.12685478. a) Correlation between altitude and the number of positive COVID-19 cases in the American continent grouped in intervals of 100 meters. b) Altitudinal distribution of the of COVID-19 positive cases in the American continent (not grouped by altitude intervals). c) Correlation between altitude and the number of positive COVID-19 cases reported below 1,000 m in the American continent. d) Correlation between altitude and the number of positive COVID-19 cases reported above 1,000 m in the American continent.

approximately at 1,000 masl and continued at higher altitudes (Fig 1b). In separate correlation analyses only considering data from altitudes above 800, 1,000, 1,500, and 2,500 masl, we confirmed this observation. No significant correlation was found for data below 1,000 masl ($p = 0.568$; r = -0.206) (Fig 1c), while a strongly significant correlation between COVID-19 incidence and altitude was obtained for data above 1,000 masl ($p<0.0001$; r = -0.675) (Fig 1d). Furthermore, considering that various factors, such as public health policies, diagnostic strategies, confinement rules, and cultural aspects, may influence the number of reported cases of COVID-19, we performed a randomized block design ANOVA test to compare the number of COVID-19 cases above and below 1,000 m. In line with our previous results, this test revealed a significant difference between the number of COVID-19 cases observed above versus below 1,000 masl (RBD-ANOVA F = 5,273; df = 45; $p = 0.022$;). Such difference is clearly evidenced when the number of cases normalized by population density is plotted showing its geographical and altitudinal distribution (Fig 2).

In the next step, we wanted to test whether the negative correlation found between altitude and the incidence of COVID-19 the American continent can be independently reproduced in each of the 14 American countries (of 23 analyzed) that reported cases over 1,000 masl. Our results showed significant negative correlations in 9 of 14 countries (Table 1) (S1 Fig): Argentina, Brazil, Canada, Colombia, Costa Rica, Ecuador, Mexico, Peru, and USA.

Taken together, these findings show a significant decrease in the incidence of COVID-19 starting above 1,000 m of altitude.

## 3.2. The transmission rate of SARS-CoV-2 is lower in the highlands compared to lowlands

To investigate whether the transmission rate of SARS-CoV-2 differs between highlands (>1,000 masl) and lowlands (<1,000 masl), we used SEIR epidemiological models. We estimated the probability of transmission of the disease (a parameter of the SEIR model) (Table 2) for those countries that applied similar strong and early quarantines and provided daily epidemiological data at state/province/departamento level: Argentina, Bolivia, Colombia, Ecuador, and Peru. We observed that compared to lowlands (Argentina = 3.73%, Bolivia = 3.57%, Ecuador = 3.88%, Peru = 3.90%), lower values of probability of transmission at highlands (Argentina = 2.04%, Bolivia = 2.69%, Ecuador = 3.44%, Peru = 2.75%) modelled data better-fitting with the real epidemiological curves. Conversely, Colombia showed lower transmission of the disease in lowlands (3.36%) compared to highlands (3.51%) (Fig 3). Overall, these results strongly support the hypothesis of decreased SARS-CoV-2 virulence in highlands compared to lowlands.

Of note, for Colombia, the "probability of transmission" value in the highlands is higher than in the lowlands. This result may be explained by the fact that 60% of its population (~ 30 million inhabitants) is settled above 1,000 m altitude, in the most densely populated areas. This causes the requirement of a higher infection probability value in our SEIR model to fit the real data, thus approaching the values of the transmission rate between highlands and lowlands.

Finally, we noticed that, except for Colombia, the "basic reproduction number" (the number of secondary cases generated by an infected individual) was consistently lower in the highland population that in the lowland population of the studied countries (Table 2).

## 3.3. The severity of COVID-19 is reduced in highlands compared to lowlands

Classically, the estimation of the severity of a disease is performed by a comparison between recovery rates and infection mortality rates (IFR). However, since this information is still

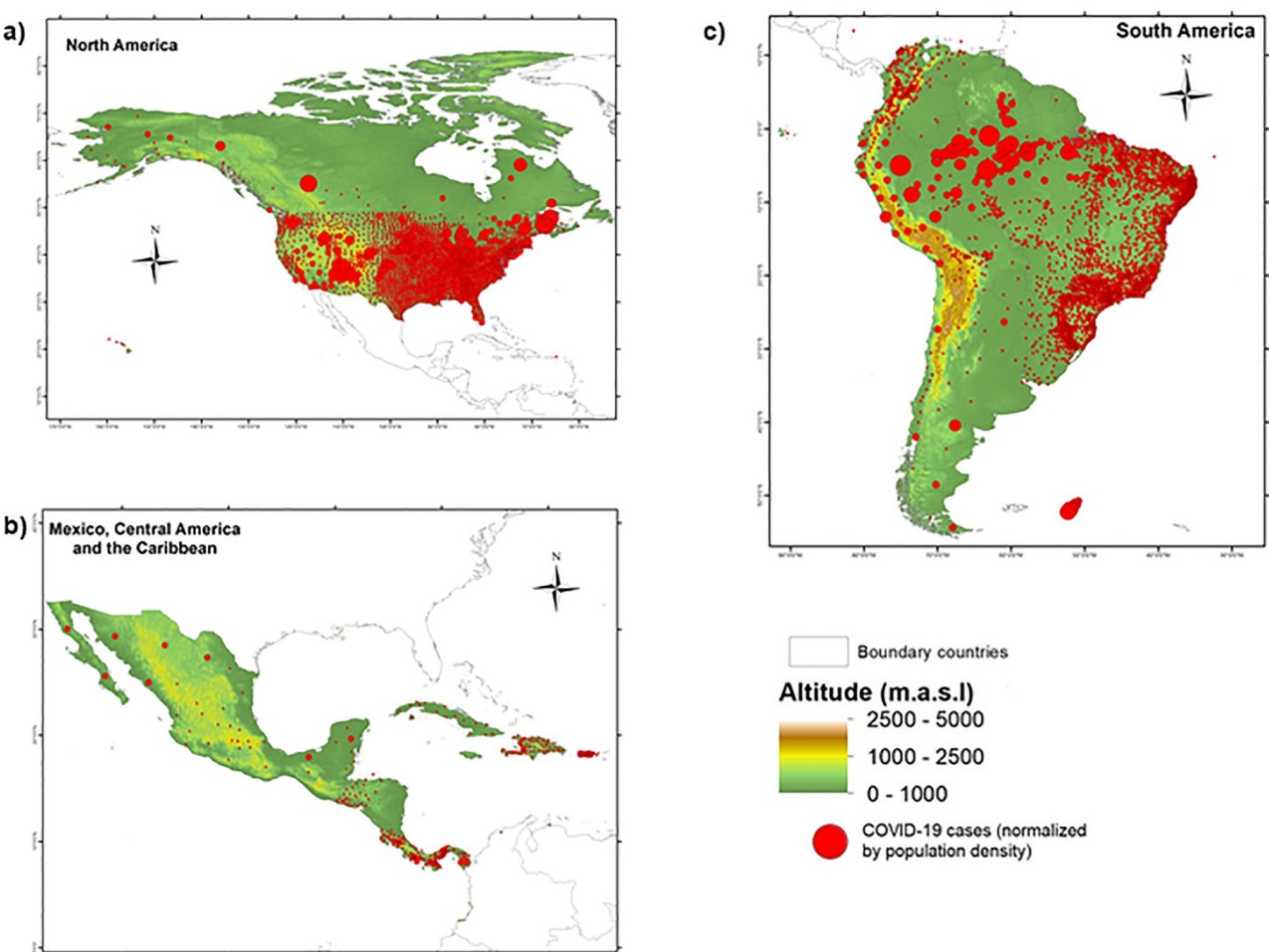

**Fig 2. Geographic and altitudinal distribution COVID-19 in a) North America, b) Central America and c) South America.** Red circles represent COVID-19 positive cases; the radius of the circle is relative to the normalized number of cases (cases/population density) in the location. The geographical coordinates and epidemiological data were retrieved on May 23, 2020 as described in the methods section. The final database used to produce these maps is available at https://doi.org/10.6084/m9.figshare.12685478. Maps for the 23 studied countries are available at https://doi.org/10.6084/m9.figshare.12685664.v1.

limited for COVID-19, we used a theoretical rationalization approach to estimate this parameter. We evaluated the differences in the percentage of recovered patients (from the total reported cases) and in the death-to-case ratio (deaths/total reported cases) as indicators of the recovery rate and the IFR for Argentina, Bolivia, Colombia, Ecuador, and Perú. We found a significantly higher percentage of recovered patients in the highlands (Wilcoxon signed-rank test $p = 0.031$) versus lowlands, suggesting a higher recovery rate in the highlands versus lowlands. On the other hand, our results did not show significant differences of death-to-case ratio between the highlands and the lowlands (Table 3). This may occur only in the case in which the number of undiagnosed cases (asymptomatic + undiagnosed symptomatic) is not similar between the highlands and lowlands. Indeed, our theoretical approach for the assessment of undiagnosed cases showed approximately 76% of undiagnosed cases in the highlands and 73% of undiagnosed cases in the lowlands (1.04 ± 0.12-fold higher) (Table 4).

**Table 1. Correlation between the altitude and the incidence of COVID-19 in American countries.**

| Country | Max.–Min. altitude w/reported cases (masl) | Pearson r | Pearson correlation *p* |
|---|---|---|---|
| Argentina | 20–3,211 | -0.773 | **0.003** |
| Belize | 0–654 | 0.4678 | 0.690 |
| Bolivia | 109–4,176 | -0.1896 | 0.343 |
| Brazil | 0–1,615 | -0.8885 | **<0.0001** |
| Canada | 1–2,115 | -0.4606 | **0.012** |
| Chile | 351–3,169 | 0.2416 | 0.405 |
| Colombia | 0–3,629 | -0.6135 | **<0.0001** |
| Costa Rica | 0–2,077 | -0.617 | **0.004** |
| Cuba | 18–315 | -0.9614 | 0.673 |
| Dominican Republic | 15–1,113 | -0.374 | 0.321 |
| Ecuador | 4–3,976 | -0.8045 | **<0.0001** |
| El Salvador | 3–788 | 0.07365 | 0.862 |
| French Guyana | 187–187 | Not enough data | Not enough data |
| Haiti | 0–1,499 | -0.5594 | 0.149 |
| Honduras | 4–1,689 | -0.282 | 0.374 |
| Mexico | 1–2,593 | -0.4636 | **0.002** |
| Panama | 0–1,402 | -0.393 | 0.184 |
| Paraguay | 61–501 | -0.7939 | 0.109 |
| Peru | 7–4,765 | -0.5336 | **0.027** |
| Puerto Rico | 0–846 | 0.3873 | 0.3432 |
| Uruguay | 27–163 | Not enough data | Not enough data |
| USA | 0–3,510 | -0.7597 | **<0.0001** |
| Venezuela | 9–1,907 | -0.4057 | 0.1907 |

Although, these values are theoretical (and should be interpreted carefully), they suggest that the severity of COVID-19 is lower at highlands compared to lowlands.

## 4. Discussion

By expanding our analysis to 23 countries of the American continent, this work clearly demonstrates our previous observations suggesting that the virulence of SARS-CoV-2 decreases significantly with altitude. Indeed, in a previous study we reported that global data analysis (that included detailed information from the Tibetan Autonomous Region of China, Bolivia, and Ecuador) suggested that the incidence of COVID-19 decreases significantly at high altitude (2,500 masl) [16]. While subsequent reports from other teams confirmed this observation [17–19, 38], in this new study we appropriately normalized the COVID-19 data by population

**Table 2. COVID-19 probability of transmission and basic reproduction numbers ($R_0$) for highland and lowland populations.**

| | Lowlands (<1,000 masl) | | | Highlands (>1,000 masl) | | |
|---|---|---|---|---|---|---|
| | Probability of transmission (%) | Confidence interval | $R_0$ | Probability of transmission (%) | Confidence interval | $R_0$ |
| **Argentina** | 3.731 | (3.728, 3.733) | 2.29 | 2.038 | (2.028, 2.048) | 1.25 |
| **Bolivia** | 3.575 | (3.572, 3.579) | 2.17 | 2.689 | (2.683, 2.696) | 1.63 |
| **Colombia** | 3.357 | (3.355, 3.36) | 2.27 | 3.511 | (3.508, 3.513) | 2.38 |
| **Ecuador** | 3.878 | (3.877, 3.88) | 2.53 | 3.443 | (3.441, 3.445) | 2.25 |
| **Peru** | 3.9046 | (3.9041, 3.9051) | 3.32 | 2.752 | (2.75, 2.754) | 2.34 |

## The transmission rate of SARS-CoV-2 is decreased above 1,000m of altitude

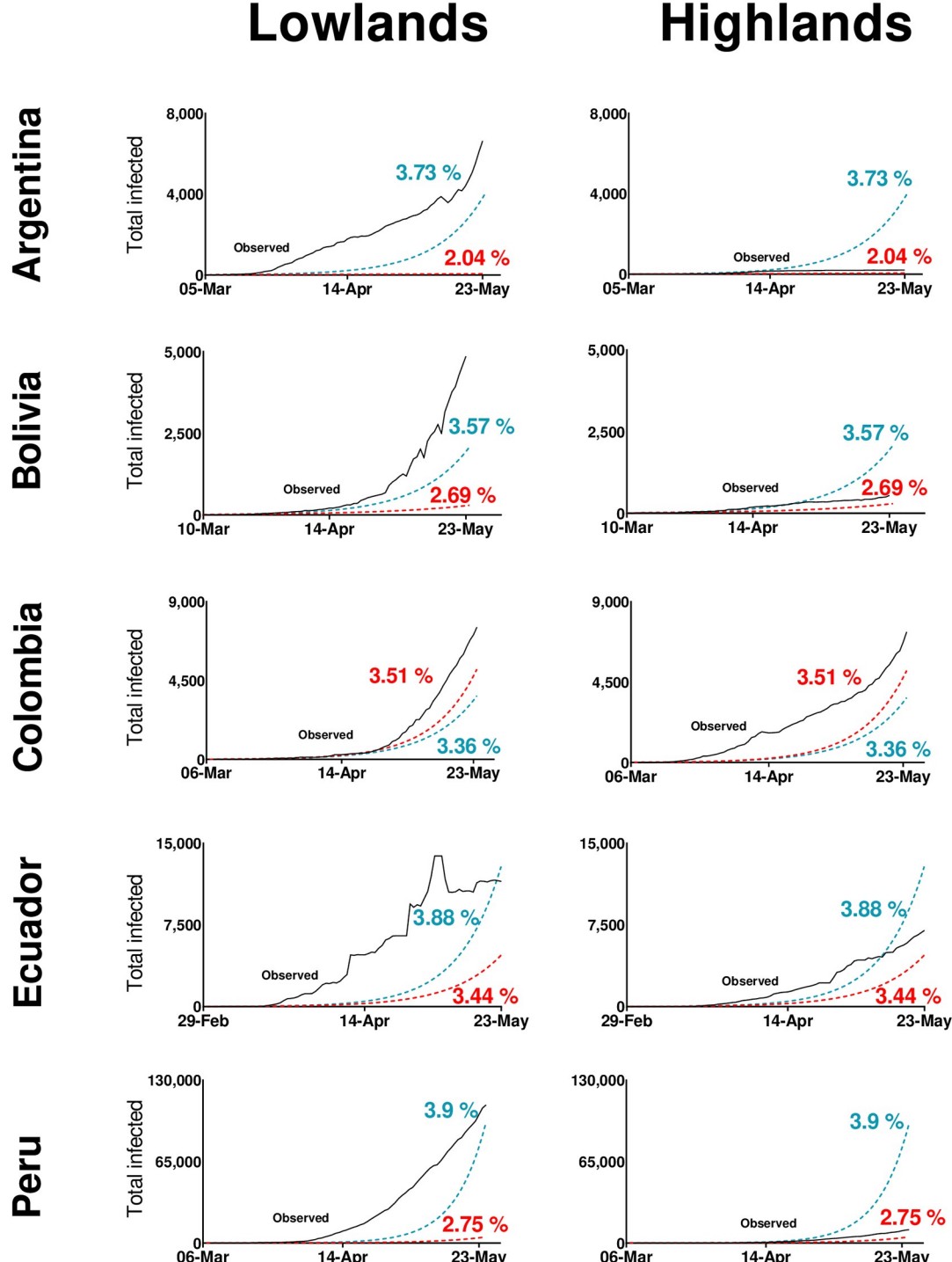

**Fig 3. Effect of the probability of transmission of the disease in the epidemiological pattern of COVID-19 in lowlands and highlands in Argentina, Bolivia, Colombia, Ecuador, and Peru.** In these countries, strict early quarantines were applied and daily epidemiological data at state/province/departamento were available by May 23, 2020. For each country, the black lines show the observed cases. The red dotted lines represent the modeled data using the optimal value of "probability of transmission" estimated for highland populations. The blue dotted lines represent the modeled data using the optimal value of "probability of transmission" for lowland populations. Percentage values are the "probability of transmission" values used for calculating the line of the same color.

**Table 3. Percentage of recovered and death rates in COVID-19 patients.**

| Country | Percentage of Recovered Patients | | Death-to-case-ratio | |
|---|---|---|---|---|
| | Highlands (%) | Lowlands (%) | Highlands (%) | Lowlands (%) |
| Argentina | 57.57 | 36.05 | 4.66 | 4.95 |
| Bolivia | 35.73 | 5.62 | 5.79 | 3.84 |
| Colombia | 4.05 | 3.46 | 36.42 | 15.7 |
| Ecuador | 9.1 | 2.38 | 9.56 | 36.0 |
| Peru | 25.77 | 8.22 | 3.5 | 3.70 |

density. Remarkably, our results show that a clear turning point in the incidence of the disease occurs at 1,000 masl. Moreover, independent examination from all the American countries that reported cases at more than 1,000 masl (14 out of 23) confirmed this result, except for four nations: Bolivia, in which the low number of cases reported for moderate-altitude regions (1,000–2,500 masl) masked a possible correlation; Chile, in which its long and narrow geography (flanked by the Andean mountains on the east and the Pacific Ocean) does not allow obtaining precise altitude values for COVID-19 cases; and Dominican Republic, Haiti, and Panamá, in which the highest altitude with reported cases is barely over 1,000 masl. Therefore, these data may have a low weight in the statistical evaluation. In agreement with our results, a negative correlation between altitude and the incidence of COVID-19 has been found in Colombia by August 1st, 2020 [39], also the excess mortality, indicator of mortality due to COVID-19, reduces while altitude increases in Peru [40].

To better understand these results, we performed additional theoretical approaches to determine the capability of the virus to pass from one host to another (the transmission rate [or the probability of transmission] of the virus). Our results show that this parameter decreases significantly in the highlands compared to lowlands, suggesting that environmental factors influence the virulence of SARS-CoV-2 at above 1,000 m. Accordingly, reduced infection rates as well as prevalence and case fatality ratios of COVID-19 were found in high-altitude populations compared with lowland populations in Peru as for June and July 2020 correspondingly [41, 42]. Indeed, as altitude increases, the environment is characterized by more drastic changes in temperature between night and day, higher air dryness, and higher levels of ultraviolet (UV) light radiation [43]. In particular, UV light radiation was suggested to be an important natural sanitizer at altitude that may shorten the half-life of any given virus [44–46]. In addition, the solar radiation is also more intense at altitude, and a recent study reported that this factor may be a key factor leading to the deactivation of the virus [47, 48]. Taken together, these factors may lead to a gradual reduction of the "survival" and "virulence" capacity of the virus as altitude progresses. Finally, the size of the virus inoculum in the air

**Table 4. Estimated percentage of undiagnosed COVID-19 cases in five American countries.**

| Country | Highlands (%) | Lowlands (%) |
|---|---|---|
| Argentina | 69.2 | 64.4 |
| Bolivia | 74.8 | 61.9 |
| Colombia | 95.5 | 88.5 |
| Ecuador | 85.5 | 96.6 |
| Peru | 58.4 | 56.0 |
| **Mean** | 75.8 | 73.5 |
| **S.D.** | 15.9 | 17.9 |

should gradually decrease as the barometric pressure decreases and the distance among air molecules increases.

Apart from environmental factors that may decrease the transmission capacity of the virus, our results also suggest that physiological mechanisms associated with a prolonged exposure to barometrical hypoxia help to decrease the severity of the infection. Thus far, two factors have been suggested that may be involved in this phenomenon: A decreased expression of the virus's gateway to the body, the angiotensin converting enzyme 2 (ACE2) [16], and an increased level of erythropoietin (EPO) [49]. Stabilization of the hypoxia inducible factor 1 alpha (HIF1-a—a master regulator of the response to hypoxia) may lead the regulation of both parameters. Indeed, it was shown that exposure of human pulmonary artery smooth muscle cells (hPASMC) to hypoxia markedly decreases the expression of ACE2 [50]. Similar results were obtained in heart cells of Sprague Dawley rats after 28 days exposure to conditions equivalent to 10% $O_2$ hypoxia [51]. Furthermore, it was shown that changes in oxygen availability as small as 2% are sufficient to induce HIF activation in many tissues [52, 53]. In the same direction, HIF is the main booster of EPO production in the kidney and other tissues, and it was observed that the HIF-related oxygen sensing mechanism accurately senses altitude differences of 300 m even in low to moderate altitudes [54]. While EPO is the central factor leading the stimulation of red blood cells [55, 56], EPO also promotes adaptive cellular responses to hypoxic challenges and tissue-damaging insults in nonhematopoietic tissues [57]. As such, in the infectious context induced by the SARS-CoV-2 virus, since EPO may help improve oxygen delivery to the tissues, it may also protect other tissues from a multiple organ dysfunction and inflammation [58, 59], thus reducing the severity and fatality rate of the disease.

Finally, the limitations of our study include possible failures and delays in reporting cases as well as that our analysis does not consider the effect of different risk groups such as age, sex, comorbidities, and the possibility of reinfection. These factors are potentially important [60] and should be considered for future studies should enough information in this regard becomes available.

In conclusion, epidemiological analyses of the present work strongly suggest that the incidence of COVID-19 significantly decreases with altitude with a turning point at 1,000 masl. This effect seems to be related both to a decrease in the transmission capacity of the virus and to the physiological characteristics of altitude residents. Finally, our results suggest that knowledge of the mechanisms of physiological acclimatization to hypoxia may help to better understand the viral nature of SARS-CoV-2 as well as facilitate the discovery of new treatment strategies.

## Supporting information

**S1 Table. Capitals of American countries located above 1,000 m above sea level.**
(PDF)

**S2 Table. Sources of epidemiological data.**
(PDF)

**S1 Appendix. Equations and parameters used in the SEIR model.**
(PDF)

**S1 Fig. Effect of altitude on the incidence of COVID-19 cases in American countries.** Epidemiological data were retrieved on May 23. Data on population density were extracted from the dataset created by CIESIN [22] or the corresponding country's national statistics institute on May 23. Data were normalized by the population density of the same location, adjusted by calculating the natural logarithm (ln) of each value, and summed in intervals of 100 m of

elevation. Raw, normalized, and adjusted data are available at https://doi.org/10.6084/m9. figshare.12685478.
(PDF)

## Acknowledgments

The authors wish to acknowledge Dr Gonzalo Leonardini for his advice and assistance in the mathematical evaluation of our models, the platform CovidBot Peru for giving us access to their database of systematized daily epidemiological information from Peru, and Teresa Roncal for her inputs and thoughts on the redaction of the manuscript.

## Author Contributions

**Conceptualization:** Christian Arias-Reyes, Jorge Soliz.

**Data curation:** Christian Arias-Reyes, Favio Carvajal-Rodriguez, Liliana Poma-Machicao, Fernanda Aliaga-Raduán.

**Formal analysis:** Christian Arias-Reyes, Favio Carvajal-Rodriguez.

**Investigation:** Christian Arias-Reyes, Liliana Poma-Machicao, Fernanda Aliaga-Raduán, Danuzia A. Marques.

**Methodology:** Christian Arias-Reyes, Favio Carvajal-Rodriguez.

**Supervision:** Roberto Alfonso Accinelli, Edith M. Schneider-Gasser, Gustavo Zubieta-Calleja, Mathias Dutschmann, Jorge Soliz.

**Validation:** Christian Arias-Reyes, Danuzia A. Marques, Natalia Zubieta-DeUrioste, Roberto Alfonso Accinelli, Edith M. Schneider-Gasser, Gustavo Zubieta-Calleja, Mathias Dutschmann, Jorge Soliz.

**Visualization:** Favio Carvajal-Rodriguez.

**Writing – original draft:** Christian Arias-Reyes, Liliana Poma-Machicao, Fernanda Aliaga-Raduán, Danuzia A. Marques, Jorge Soliz.

**Writing – review & editing:** Christian Arias-Reyes, Natalia Zubieta-DeUrioste, Roberto Alfonso Accinelli, Edith M. Schneider-Gasser, Gustavo Zubieta-Calleja, Mathias Dutschmann, Jorge Soliz.

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
