## [Decision Letter · Decision Letter 0]

19 Oct 2020

PONE-D-20-23585

Decreased incidence, virus transmission capacity, and severity of COVID-19 at altitude on the American continent

PLOS ONE

Dear Dr. Soliz,

Thank you for submitting your manuscript to PLOS ONE. After careful consideration, we feel that it has merit but does not fully meet PLOS ONE’s publication criteria as it currently stands. Therefore, we invite you to submit a revised version of the manuscript that addresses the points raised during the review process.

You will see that one of the reviewers had major concerns about the methodological approach in your study. You may decide to revise parts of the study and/or to justify your original approach. In any case, I would appreciate to see all the reviewer's concerns addressed in the rebuttal letter and where possible, clarified in the manuscript as well. With this decision of "major revision", we want to give you the opportunity to write a rebuttal. I hope you understand that at this point, we cannot guarantee that we will accept the revised manuscript.

We look forward to receiving your revised manuscript.

Kind regards,

Kristien Verdonck

Academic Editor

PLOS ONE

Journal Requirements:

2. In ethics statement in the manuscript and in the online submission form, please provide additional information about the database used in your retrospective study. Specifically, please ensure that you have discussed whether all data were fully anonymized before you accessed them and/or whether the IRB or ethics committee waived the requirement for informed consent. If patients provided informed written consent to have their data used in research, please include this information.

3. Please ensure that you refer to Figure 3 in your text as, if accepted, production will need this reference to link the reader to the figure.

4.We note that [Figure(s) 3] in your submission contain [map/satellite] images which may be copyrighted. All PLOS content is published under the Creative Commons Attribution License (CC BY 4.0), which means that the manuscript, images, and Supporting Information files will be freely available online, and any third party is permitted to access, download, copy, distribute, and use these materials in any way, even commercially, with proper attribution. For these reasons, we cannot publish previously copyrighted maps or satellite images created using proprietary data, such as Google software (Google Maps, Street View, and Earth). For more information, see our copyright guidelines: http://journals.plos.org/plosone/s/licenses-and-copyright.

1.    You may seek permission from the original copyright holder of Figure(s) [3] to publish the content specifically under the CC BY 4.0 license. 

5. We note you have included a table to which you do not refer in the text of your manuscript. Please ensure that you refer to Table 1 in your text; if accepted, production will need this reference to link the reader to the Table.

Reviewers' comments:

Reviewer's Responses to Questions

**Comments to the Author**

1. Is the manuscript technically sound, and do the data support the conclusions?

Reviewer #1: No

Reviewer #2: Yes

2. Has the statistical analysis been performed appropriately and rigorously? 

Reviewer #1: I Don't Know

Reviewer #2: Yes

3. Have the authors made all data underlying the findings in their manuscript fully available?

Reviewer #1: Yes

Reviewer #2: Yes

4. Is the manuscript presented in an intelligible fashion and written in standard English?

Reviewer #1: Yes

Reviewer #2: Yes

5. Review Comments to the Author

Reviewer #1: The hypothesis of the article is based on an interesting observation, that infection attack rate and infection severity decrease with altitude. It is important to investigate whether that observation still stands when you would take reporting, demographic or epidemiological factors into account.

However, the methods used by the authors are not appropriate and sufficient to address the research question, and use rather unconventional parameters to investigate infection rates and disease severity, which makes it difficult to interpret the robustness of the results. One would expect a estimation of location-specific reproduction numbers or infection/disease incidence and infection or case fatality risks when comparing infection rates and disease severity. Instead, a number of rather complex parameters are calculated, without explaining why they are calculated that way. For instance, incidence is expressed as the natural logarithm of the number of reported cases divided by population density. Separately, a SEIR model is built, but it is unclear why, and what outcome measure this had to provide. Moreover, it is unclear how the number of infections has been estimated as part of the SEIR model, or if that was deducted from it. For severity, a 'death-to-case' ratio and pct recovered patients were calculated, rather than an infection fatality risk, which would have been more appropriate. Moreover, it is unclear at what stage during the outbreak these were estimated (during the exponential increase? which would overestimate the number of cases as compared to deaths), and it seems like no reporting+symptom to death time lag (delay between symptoms and death, and a delay in reporting deaths) were considered. Most importantly, potential third factors, which could importantly confound the association between altitude and incidence or severity, such are differences in population age structure between populations living in places with higher or lower altitude, are not taken into account.

For Figure 1, it is unclear why 4 figures are provided, and to what extent they differ. Are b, c and d just zooms of the first figure? why are some points which were shown in fig 1a missing in 1c?

For Figure 2, it is unclear what the percentages stand for, and what the different dashed lines stand for. The figure is described as 'effect', but it is a mere comparison of two observations.

For Figure 3, it is unclear to me why population density would not already be taken into account when calculating incidence in a conventional way. I would be very interested to know why you use a natural logarithm and divide by km2.

For Figure 4, comparing countries, stating quarantine measures were comparable, does not seem an appropriate way to answer your research question, for many reasons including some stated above (pop age structure, reporting differences, etc.)

Reviewer #2: Research Article is informative and interesting. Manuscript is also well written and presented.

The aricle presents epidemiological data as of 23rd May. Authors may add some more recent literature supporting their finding (if any!) and any other contrasting report (if any!).

6. PLOS authors have the option to publish the peer review history of their article (what does this mean?). If published, this will include your full peer review and any attached files.

Reviewer #1: No

Reviewer #2: No

---

## [Author Response · Author response to Decision Letter 0]

10 Nov 2020

We thank the Editor and the Referees for their important remarks that helped to upgrade the quality of our manuscript. We were pleased to see that the referees found this manuscript technically sound, statistically rigorous, and well written and presented. We wish to respond to your comments as follows:

JOURNAL REQUIREMENTS:

The style of the manuscript meets the style requirements of PLOS ONE.

2. In ethics statement in the manuscript and in the online submission form, please provide additional information about the database used in your retrospective study. Specifically, please ensure that you have discussed whether all data were fully anonymized before you accessed them and/or whether the IRB or ethics committee waived the requirement for informed consent. If patients provided informed written consent to have their data used in research, please include this information.

The information from the databases has been included in the manuscript and in the submission form. We confirmed that the data was completely anonymized before accessing it.

3. Please ensure that you refer to Figure 3 in your text as, if accepted, production will need this reference to link the reader to the figure.

Figures 2 and 3 were reorganized. Former Figure 3: “Less COVID-19 cases occur above 1,000 masl in the American continent” is Figure 2 in the revised version of our manuscript. The current Figure 2 was properly introduced in the text. 

The reference to former Figure 2, Figure 3 in the revised version of our manuscript: “The infection rate of SARS-CoV-2 is decreased above 1,000m of altitude” has been changed.

4. We note that [Figure(s) 3] in your submission contain [map/satellite] images which may be copyrighted. All PLOS content is published under the Creative Commons Attribution License (CC BY 4.0), which means that the manuscript, images, and Supporting Information files will be freely available online, and any third party is permitted to access, download, copy, distribute, and use these materials in any way, even commercially, with proper attribution. For these reasons, we cannot publish previously copyrighted maps or satellite images created using proprietary data, such as Google software (Google Maps, Street View, and Earth). For more information, see our copyright guidelines: http://journals.plos.org/plosone/s/licenses-and-copyright.

Figure 2 (former Figure 3) was created using QGIS 3.14 without using any copyrighted maps or satellite images.

Geographic data was obtained using the OpenCage Geocoding API which uses open data sources. A full list of data sources used by this API are listed here: https://opencagedata.com/credits

Altitude data was retrieved from Worldclim 2.0 data base, which explicitly authorizes its open use for research and related activities (https://worldclim.org/data/index.html).

Population density data was extracted from the dataset created by the Center for International Earth Science Information Network - CIESIN - Columbia University, and it is completely open for any use as stated in: 2018http://www.ciesin.org/documents/CIESINDataPolicy.pdf

Data of COVID-19 cases were obtained from the official public government sources of each country. All of them are open.

All the figures and datasets linked to the manuscript are hosted in the repository “figshare” under Creative Commons Attribution License (CC BY 4.0).

5. We note you have included a table to which you do not refer in the text of your manuscript. Please ensure that you refer to Table 1 in your text; if accepted, production will need this reference to link the reader to the Table.

Table 1 is now correctly referenced in the introduction section (pg. 4; Ln.68) of the revised version of our manuscript.

REVIEWER #1 COMMENTS

We thank this referee for his important observation and remarks that helped us improve our manuscript. We wish to respond to his comments as follows:

Comment 1. “One would expect a estimation of location-specific reproduction numbers or infection/disease incidence and infection or case fatality risks when comparing infection rates and disease severity. Instead, a number of rather complex parameters are calculated, without explaining why they are calculated that way.”

Answer. All statistical analyses in this work were performed based on classical epidemiological statistics. To do this, we were advised by the epidemiological research center of Laval University.

In brief, in this manuscript we made two types of analyses: 

1) Statistical, at population level: To test whether there is an effect of altitude over the incidence of COVID-19.

2) Epidemiological: To evaluate whether the transmissibility and severity of SARS-CoV-2 were affected in highlands.

Our results showing positive correlations between altitude and COVID-19, supported by the significant difference in COVID-19 incidence between locations above and below 1,000 masl (ANOVA), show a clear effect of altitude on COVID-19.

Next, at epidemiological level we calculated the “death-to-case ratio” and the “% of recovered patients”. These are the recommended statistical analyzes when the information treated (as the one in the current pandemic) is limited by the quality of the available data. Indeed, most epidemiological parameters calculated come from data series with timely registers, which, even to date, are not fully available for most American countries (usually, only total numbers were reported). Furthermore, the calculation of additional classical parameters, such as the “Infection fatality risk” (also known as Infection fatality rate - IFR) is not possible since IFR is defined as the risk of death among all infected individuals including those with asymptomatic and mild infections (Yang et al., 2020), and this information will be complete only when the pandemic ends. In consequence, in the section “3.4 The severity of COVID-19 is reduced in highlands compared to lowlands” we declare that due to this unfeasibility “we evaluated the differences in the percentage of recovered patients (from the total reported cases) and in the death-to-case ratio (deaths/total reported cases) as indicators of the recovery rate and the IFR” 

Finally, in the revised version of our manuscript, we have included the calculated values of the basic reproduction number (R0) for the lowlands and highlands of the five countries we analyzed in this section in Table 3. As can be seen, consistently R0 values in highlands are lower than in lowlands.

Comment 2. “For instance, incidence is expressed as the natural logarithm of the number of reported cases divided by population density.”

 Based on epidemiological statistics, the explanation for this is that:

1. Being SARS-CoV-2 a respiratory virus, it has been suggested that it is more easily transmitted between people in more densely populated places. In this work, we clearly show that there is a significant correlation between the population density and the incidence of COVID-19 (S2). Furthermore, assuming that the high-altitude settlements are less densely populated than the lowlands, it is necessary to normalize the number of cases at each location by the corresponding population density. Classically, the incidence of pathology is expressed as "number of cases per 100,000 inhabitants", however, this parameter does not reflect the effect of population density. For further explanation, please read the response to comment 9.

2. The normalization of the incidence of COVID-19 (# of cases/population density), results in very dispersed values. That is, overpopulated cities with few cases will have very small normalized values (white cells), while less dense cities with many cases will result in very high normalized values (grey cells). See the following example:

City Province/State Country/Region Cases Pop_den #Cases/Pop_den

Tibas San Jose Costa Rica 25 13463.556 0.001856864

Barra do Turvo São Paulo Brasil 5 3412.5 0.001465201

Montgomery Arkansas USA 1 97.49 0.010257105

Durham Ontario Canada 1358 1.58140 858.7327684

Lima Lima Peru 74037 272.4 271.7951542

McKinley New Mexico USA 2192 5.19 422.3506744

In this type of dispersion, a logarithmic fit of the data is recommended to facilitate its analysis. 

Comment 3. “Separately, a SEIR model is built, but it is unclear why, and what outcome measure this had to provide. Moreover, it is unclear how the number of infections has been estimated as part of the SEIR model, or if that was deducted from it.”

Answer. A better explanation of this analysis has been included in the “Results” section of the revised version of our manuscript (pg. 13-14; Ln. 271-276). 

First, we used SEIR models to replicate (mathematically) the real data reported for the lowlands (<1,000 masl) and highlands (>1,000 masl) of Argentina, Bolivia, Colombia, Ecuador, and Colombia. To do so, we calculated the number of “Susceptible”, “Exposed”, “Infected”, and “Removed” individuals (from the date the first case was reported in the corresponding country until May 23) using the theoretical parameters (initial number of infected, number of exposed subjects, contact rate, recovery rate, and the rate at which exposed individuals become infected) as described in the methods section. Next, we adjusted such parameters of the model to match the real reported numbers of “Infected” people for the highland and lowland populations separately. In doing so, in the mathematical model, we “played” with the "transmission rate" in such a way that they allow the most faithful reproduction of the epidemiological curves observed in the highlands and lowlands. For the five above mentioned countries, we found that using lower values of transmission rates reproduce better the real data for highland populations. On the contrary, higher values of transmission rates reproduced better the real data for lowland populations.

Comment 4. “For severity, a 'death-to-case' ratio and pct recovered patients were calculated, rather than an infection fatality risk, which would have been more appropriate.”

Answer. Please see the answer provided for Observation 1. 

Comment 5. “Moreover, it is unclear at what stage during the outbreak these were estimated (during the exponential increase? which would overestimate the number of cases as compared to deaths), and it seems like no reporting+symptom to death time lag (delay between symptoms and death, and a delay in reporting deaths) were considered.”

Answer. As previously mentioned, the data analyzed correspond to those collected from the date of notification of the first case (for each country) until May 23. Regarding the epidemiological analyses to which the reviewer refers, in Argentina, Bolivia, Colombia, Ecuador, and Peru this period includes early stages of the exponential increase phase in the lowlands, but not in the highlands (with exception of Colombia), where cases remained low passed May 23 (Fig 3). 

The delays between the notification of symptoms and deaths and the delay in the notification of deaths are not available, much less at the required geographical level (state/province/departamento). Therefore, it is not possible to relate these data to altitude. As far as we know, even today, these data, with such geographic resolution, are not available. Furthermore, the Pan American Health Organization (PAHO) informed us that they do not have this information in their databases either. Although the referee's observation is pertinent, it will not be possible to carry out this type of analysis until the pandemic is over and all the data is completed in detail and made available.

Comment 6. “Most importantly, potential third factors, which could importantly confound the association between altitude and incidence or severity, such are differences in population age structure between populations living in places with higher or lower altitude, are not taken into account.”

Answer. 

We thank this referee for this important observation. Indeed, we mentioned this limitation in our work in the discussion section of our manuscript (pg. 20; Ln. 419-423). It is important to note, however, that to date, with few exceptions, official COVID-19 data sources (i.e. governments, health agencies, and research institutes), only provided information on the total number of cases (cumulative), total deaths, and in some cases the total number of recoveries. So far, no institution, in any country, has provided information on more detailed epidemiological factors, such as age structure, comorbidities, or sex of those infected.

Please note that with all these limitations, we were able to evaluate and analyze the COVID-19 incidence data from more than 8,000 locations corresponding to 23 countries within the American continent. In this sense, we believe that our reports have great epidemiological value and will serve as a basis for the development of new studies, we hope that they will be more complete and detailed.

Comment 7. “For Figure 1, it is unclear why 4 figures are provided, and to what extent they differ. Are b, c and d just zooms of the first figure? why are some points which were shown in fig 1a missing in 1c?”

Answer. We thank the reviewer for this important observation. In effect, we put an incorrect graph on panel c in Figure 1. The correct graphic is now in place.

Panel a shows the correlation between altitude and the incidence of COVID-19 considering the entire altitudinal range (0 - 4,800 masl) of locations with COVID-19 cases in the American continent. In this figure, the points (open triangles) represent the summatory of the incidence every 100 meters of altitude. This graph is important because it shows an effect of altitude on the incidence of COVID-19.

Panel b shows the same data as panel a but broken down for each altitude (the data without grouping every 100 meters of altitude). This graph is important because it shows that there is a significant cut in the incidence of COVID-19 at 1,000 meters above sea level (shown by the red dotted line).

To make this cut of COVID-19 incidence at 1,000 meters more evident, we carried out the analyzes showed in panels c and d, which evidence that there is indeed no incidence of altitude up to 1,000 meters (Panel c) and that the effect of altitude begins from 1,000 meters up (Panel d). These last two graphs are important because they statistically show that the effect of altitude begins at 1,000 meters.

A clearer description of this figure was included in the corresponding legend of the corrected version of our manuscript.

Comment 8. “For Figure 2, it is unclear what the percentages stand for, and what the different dashed lines stand for. The figure is described as 'effect', but it is a mere comparison of two observations.”

Answer. 

Note that figure 2 above is the current figure 3.

The percentage values are the “transmission rate” values that are used to theoretically calculate the numbers of susceptible, exposed, infected, and removed people over time with the SEIR model for COVID-19. As can be seen in this figure, to make a representative theoretical calculation of these curves, different "transmission rate" values are necessary for the lowlands and the highlands. As such, if we use the same "transmission rate" for lowlands and highlands, the theoretically calculated graphs would not reflect the reality (solid black lines). Thus, the percentage values in blue are the "transmission rate" that is suitable for modelling the lowland data (dotted lines in blue). Instead, these values in the highland figures show how the same "transmission rate" does not model the real data from highlands. On the other hand, the percentage values in red are the "transmission rate" that is suitable for modelling the data in highlands (lines dotted in red). These graphs are important because they show that to model the highland data of COVID-19 infection, lower "transmission rates" of the virus are required than those required for modelling the data of lowlands. Biologically, this implies that the probability of transmission of the SARS-CoV-2 virus is reduced in the highlands compared to lowlands.

In the revised version of our manuscript, we include a more detailed explanation of this figure in the corresponding legend.

Comment 9. “For Figure 3, it is unclear to me why population density would not already be taken into account when calculating incidence in a conventional way. I would be very interested to know why you use a natural logarithm and divide by km2.”

Answer. Incidence, traditionally reported as number of cases/100,000, inhabitants is mathematically limited by the dividend to the number of total population in a zone, without considering the total area (km2) of such zone. For a better clarification see the following example comparing two fictitious cities:

Scenario 1: Only the number cases is different between the two cities.

Scenario 2: Only the population is different between the two cities (and this changes the population density).

Scenario 3. Only the area is different between the two cities (and this changes the population density).

Scenario 4. The area and the number of cases are different between the two cities.

 Population Area (km2) Pop. Density (people/km2) COVID-19 cases Cases/100,000 people Cases/pop. Dens.

Scenario 1 City A 500,000 100 5000 50 10 0.01

 City B 500,000 100 5000 10 2 0.002

Scenario 2 City A 500,000 100 5000 50 10 0.01

 City B 250,000 100 2500 50 20 0.02

Scenario 3 City A 500,000 100 5000 50 10 0.01

 City B 500,000 50 10000 50 10 0.005

Scenario 4 City A 500,000 100 5000 10 2 0.002

 City B 500,000 50 10000 50 10 0.005

In scenarios 1 and 2, both ways to calculate incidence (Cases/100,000 people and Cases/population density) are equivalent. However, in scenarios 3 and 4, when the population density is different between the two cities due to changes in the area, normalizing the number of cases by population density results in a higher value of incidence (in comparison with the other method), thus, revealing locations where the small number of COVID-19 cases is related with low population densities. Such situation has been suggested to happen in rural settlements (particularly in high altitudes), where people live far away from each other.

Regarding the logarithmization, please see the answer 2.

Comment 10. “For Figure 4, comparing countries, stating quarantine measures were comparable, does not seem an appropriate way to answer your research question, for many reasons including some stated above (pop age structure, reporting differences, etc.)”

Answer. Figure 4 shows the number of infected people estimated for highland populations of Argentina, Bolivia, Colombia, Ecuador, and Peru, in a scenario in which quarantines would not be applied in these countries. Figure 4 presents the real (reported) data in the black-dotted line, the blue line represents the data modelled (SEIR models) to emulate the real data and the red line represents the modelled data using a higher value of “frequency of interaction” (a parameter of SEIR models) to simulate the absence of a quarantine. The values of frequency of interaction used to calculate the blue and red lines are detailed in the methods section.

As stated in the main text (pg. 20; Ln. 408-414), the intention of this analysis is to show that social isolation measures are crucial to reduce the number of infected people regardless of altitude. This is important because the readers of this report could interpret our results as that quarantine and social isolation measures, especially in highlands, are not necessary to decrease the transmission of the virus.

Moreover, our modelled data (blue lines) emulates well the numbers of infected people for each of the five countries analyzed regardless of the omission of more detailed epidemiological parameters as those mentioned by the reviewer. This shows that all those parameters, although remarkable, are not determinant to reach the conclusions obtained in this work. In any case, as mentioned above, such additional parameters are not available, especially at the level of geographic resolution required for this work, and it is possible that these data will be available for analysis one or two years after the end of the pandemic.

REVIEWER #2 COMMENTS

We were pleased to see that this referee stated that our manuscript is informative, interesting, and well written and presented.

Comment 1. The article presents epidemiological data as of 23rd May. Authors may add some more recent literature supporting their finding (if any!) and any other contrasting report (if any!)

Answer. The revised version of our manuscript includes, in the discussion section, the references of recently published works (after the initial presentation of this manuscript).

BIBLIOGRAPHY CITED

Yang, W., Kandula, S., Huynh, M., Greene, S. K., Van Wye, G., Li, W., . . . Olson, D. (2020). Estimating the infection-fatality risk of SARS-CoV-2 in New York City during the spring 2020 pandemic wave: a model-based analysis. The Lancet Infectious Diseases.

---

## [Decision Letter · Decision Letter 1]

13 Jan 2021

PONE-D-20-23585R1

Decreased incidence, virus transmission capacity, and severity of COVID-19 at altitude on the American continent

PLOS ONE

Dear Dr. Soliz,

Thank you for submitting your manuscript to PLOS ONE. After careful consideration, we feel that it has merit but does not fully meet PLOS ONE’s publication criteria as it currently stands. Therefore, we invite you to submit a revised version of the manuscript that addresses the points raised during the review process.

We look forward to receiving your revised manuscript.

Kind regards,

Kristien Verdonck

Academic Editor

PLOS ONE

Reviewers' comments:

Reviewer's Responses to Questions

**Comments to the Author**

1. If the authors have adequately addressed your comments raised in a previous round of review and you feel that this manuscript is now acceptable for publication, you may indicate that here to bypass the “Comments to the Author” section, enter your conflict of interest statement in the “Confidential to Editor” section, and submit your "Accept" recommendation.

Reviewer #3: (No Response)

2. Is the manuscript technically sound, and do the data support the conclusions?

Reviewer #3: No

3. Has the statistical analysis been performed appropriately and rigorously? 

Reviewer #3: No

4. Have the authors made all data underlying the findings in their manuscript fully available?

Reviewer #3: Yes

5. Is the manuscript presented in an intelligible fashion and written in standard English?

Reviewer #3: No

6. Review Comments to the Author

Reviewer #3: Peer review: Decreased incidence, virus transmission capacity, and severity of COVID-19 at altitude on the American continent

Summary

This study aims to evaluate the impact of altitude on the manifestation of SARS-CoV-2. In particular, the correlation between altitude and incidence of COVID-19, it’s severity, and the transmissibility of SARS-CoV-2. The authors focus on the countries of the American continent. Overall, the authors are transparent about the methods used and their thought process. However, the manuscript can do with some good editing. More specifically, without wanting to come across mean or rude, it appears the manuscript is not written by someone with a statistical background and can do with more clarification (see below for more detail) as well as a rewrite, e.g. pp 6 line 107-115, terms like “the analysed variable” are not conventional terms to use, rather something along the lines of “dependent variable”. Also, sections in the results are better suited in the discussion and/or methods. Moreover, the manuscript comprises quite some repetition in methods used (e.g. normalised and logarithmatised is repeated many times unnecessarily). Finally, and perhaps more importantly, I have some doubts about the methods employed and interpretation of the results, among which the methods used to assess the respective transmissibility in the ‘highlands’ vs ‘lowlands’.

Major comments

- The authors use the Pearson correlation coefficient to assess among others the linear relationship between altitude and incidence rates and incidence and population density.

o First of all, this test is valid when both variables of concern are normally distributed. Could the authors please confirm whether they assessed normality in their variable distributions? Otherwise a non-parametric test might be more suited.

o Secondly, R2 is listed along side the estimated pearson’s correlation. This could be me, but I would say reporting pearson’s r is more common. Can the authors confirm what is reported is the R2 and why? Now more importantly, the authors report on the significance of their correlation between COVID-19 incidence and population density. Although I have nothing against normalising the result by population density, I doubt relying on merely a significant p-value with such a low R2 provides the right ‘prove’ to do so (this might relate to a high sample size, but as listed in the minor comments, it is a bit unclear to me which test is fitted to which data). I think this is also confirmed by the high variance observed in the correlation between these two variables in Figure S2. Perhaps better to explain rational for normalising incidence by population density in the methods and leave out 3.1.

o I find the authors conclusions more concerning for table 2, where significant p values go alongside with a wide range of R2 values. Explanation is in part covered in the discussion section pp 18, but this is for the countries where no correlation is found. I think this could be done more elaborate, among which how quality of passive surveillance could affect the findings in terms of strength of the correlation.

- The authors are speaking of “the SEIR” model, but in fact, SEIR model structures can involve a multitude of assumptions and parameters encompassing these assumptions. As a result, I have some difficulty assessing the validity of the findings regarding the evaluation of the virus transmission rates.

o Therefore, first of all “an SEIR model structure” would be more appropriate on pp6 line 122.

o Also, a listing of the differential equations either in the methods section or the supplementary material would be useful to provide the reader transparency in what model parameters and assumptions were incorporated. E.g. what is assumed regarding the transmission rate of asymptomatic individuals? Non infectious at all? What fraction of the population is assumed to remain asymptomatic?

o Also, it is good practice to list the parameters values used as well as their source. Therefore, I would argue for a table in the supplementary material.

o Moreover, the interaction frequency needs explaining. A more commonly used term I think is “contact rate”. The authors refer to beta as their contact rate, but this would more commonly be coined “the effective contact rate” (which can be denoted as a product of contact rate and probability of infection upon contact).

- More importantly, heterogeneity in transmission of SARS-CoV-2 has been outspoken (e.g. https://wellcomeopenresearch.org/articles/5-67/v3, https://doi.org/10.1101/2020.08.09.20171132 ), which has among others been associated to heterogeneities in age-specific contact patterns. Therefore, estimating an R0 based on an average contact rate, probability of transmission as well as infection period is not a very accurate representation of SARS-CoV-2 dynamics. Importantly, quantifying differences between country-level altitude, by “playing around” with the transmission rate does not come across methodologically sound and could, at minimum, do with a formal fitting procedure.

- Moreover, in my opinion the differences between transmission rates in the high and lowlands are hardly convincing. A deterministic model is used, and confidence intervals are missing, but my suspicion is that these would highly overlap.

- What I am also not certain about (but I might have missed it), do the authors vary the “frequency of interaction rate” between high and lowlands? As the SEIR model is ‘fitted’ to raw incidence, one would expect different frequency of interaction rates in higher than in lower densely populated areas. In a dynamic transmission model where beta, the effective contact rate, is a product of the frequency of interaction and the probability of infection upon contact, the ‘fitted’ value for the latter will be correlated to the value used for the former. This needs clarification.

- In conclusion, I feel rather uncomfortable by the SEIR model used and validity of the findings reported on the transmission rates and R0 estimates between highlands vs lowlands. I doubt these findings should be part of the manuscript.

- Estimate for severity is based on the recovered to case ratio and recovery rates. For clarification, what do the deaths represent in the death to case ratio? The national reported covid deaths? Excess mortality? As I am not entirely sure why there should be such a gap between 1- fraction of of recovered patients (recovered/total cases) and death to case ratio (deaths/total cases).

- Regardless, both are very likely sensitive to underreporting (i.e. the higher underreporting, the higher the death-to-case ratio). This is the reason why the authors evaluate the underreporting of cases in high and lowlands. The authors conclude that the non-significant differences observed in death-to-case ratio between high and lowlands could be explained by differences in in undiagnosed cases (76% for highlands vs 73% for lowlands). Estimating confidence intervals based on mean and SD will, I think, show that these estimates overlap, i.e. revealing that this might not necessarily explain the non-observed difference. Even if there was a true difference in underreporting between high and lowlands, why would one expect a difference in underreporting based on altitude? Please clarify and what could be an alternative explanation. In particular why there is a difference in %recovered but not in death to case ratio (but as stated, I don’t fully understand the difference).

- Also methods pp 7 lines 149 – 151: COVID severity: These seem to be based on national estimates (npairs = 5). This while the estimates for the correlation between cases and altitude and case numbers are based on a mixture of national, regional and local level data (see minor comments). Why the difference? To what extend does it make sense to use national level data here, why not, similar to the analyses correlating altitude with incidence, on a more granular level?

- Discussion pp 19: This seems to elaborate quite far on what is covered in the manuscript. Virus transmission capacity under different altitudes I think is what the authors want to elude to, but it now comes across somewhat as a self-citation exercise and covering a topic the authors is probably familiar with. Suggest to cut and/or shorten.

Minor Comments

- Some clarifications/editing in the methods would be useful, i.e.:

o Methods pp 5 line 105: what do the n=51 represent? Is this a mixture of observations on city or province or state or country or departmental level?

o Similar for pp 6 lines 109-110, what do the n’s represent here? It seems country-level datapoints, but as the block variable concerns ‘country’ there must be multiple observations from within the countries included. Moreover, these are estimates all across the whole world, not only the American continent? And what about the n’s listed in line 111-115 for those countries in the American continent? Please clarify.

o For the non-statistical reader, it would be good to refer to “the block variable” in a more intuitive way.

o pp 6 line 107-115, terms like “the analysed variable” are not conventional terms to use, rather something along the lines of “dependent variable”.

o “organised by intervals of 100m of altitude”. ‘Categorised’ or ‘grouped by’ might be preferred.

o Pp 6 line 122: The SEIR should be An SEIR. There is not just one SEIR model.

o Overall, the methods section could do with a revision from a statistician/epidemiologist.

- There are certain sections in the result section, that are better suited in the discussion, as these provide interpretation of the results in the context of existing evidence, i.e.:

o Results pp 10 line 196 – 197. This should be moved to the discussion.

o Results pp 12 lines 258-261: This should be moved to discussion.

- Also, no need to repeat again that cases were normalised by population density. The same holds for pp 10 lines 203-204. The methods clearly describe what scale was assumed for the correlations, no need to list this again.

- Methods pp 10 line 207: I think a word is missing in this sentence, but also I don’t fully understand what is meant. I think that beyond 1000 masl, a correlation between altitude and COVID-19 incidence is apparent (and not below) but this is not what it reads.

- I also don’t understand what the authors have done when they state “repeated correlation analyses performed at altitudes above 800, 1000 etc.”

- Methods pp 11 lines 219 – 221: Move to methods.

- Figure 1B legend: add “above 1000m”?

- Fig 2: It might be the resolution, but I fail to see the blue circles. Should these be ‘red circles’?

-

Typos

- Introduction pp4 line 76: Appears to have referencing non-consistent with the remainder of the article

- Introduction: It’s not very common to include tables in the introduction. Is this table really needed? Or can it be moved to Supplementary material?

- Methods pp 8 lines 160-165: Sentence is not correct.

- Methods pp 8 line 167: Ref 28 (worldometer) the correct reference?

-

7. PLOS authors have the option to publish the peer review history of their article (what does this mean?). If published, this will include your full peer review and any attached files.

Reviewer #3: **Yes: **Esther Van Kleef

---

## [Author Response · Author response to Decision Letter 1]

26 Feb 2021

We thank the Referee for her important remarks that helped to upgrade the quality of our manuscript. We were pleased to see that the referee found our manuscript to be transparent about our methods and thinking process. We wish to respond to your comments as follows:

REVIEWER #3 COMMENTS 

 SUMMARY

This study aims to evaluate the impact of altitude on the manifestation of SARS-CoV-2. In particular, the correlation between altitude and incidence of COVID-19, it’s severity, and the transmissibility of SARS-CoV-2. The authors focus on the countries of the American continent. Overall, the authors are transparent about the methods used and their thought process. However, the manuscript can do with some good editing. More specifically, without wanting to come across mean or rude, it appears the manuscript is not written by someone with a statistical background and can do with more clarification (see below for more detail) as well as a rewrite, e.g. pp 6 line 107-115, terms like “the analysed variable” are not conventional terms to use, rather something along the lines of “dependent variable”. Also, sections in the results are better suited in the discussion and/or methods. Moreover, the manuscript comprises quite some repetition in methods used (e.g. normalised and logarithmatised is repeated many times unnecessarily). Finally, and perhaps more importantly, I have some doubts about the methods employed and interpretation of the results, among which the methods used to assess the respective transmissibility in the ‘highlands’ vs ‘lowlands’.

 Pp6 lines 114-115: The text has been changed, now it reads: “The dependent variable was…”

 The text “normalised and logarithmized” was removed from the manuscript in several parts to avoid repetition. Also, the following text has been added in the Methods section:

Pp 5 lines 104-105: “These data (referred as the number COVID-19 cases) were used for all the analyses unless stated otherwise.”

 Some sentences in the results sections have been moved to the methods section according to the suggestions of the reviewer (see minor comments).

 Regarding the methods employed to assess the transmissibility, please see the answer to the comment 2.5.

 MAJOR COMMENTS

 The authors use the Pearson correlation coefficient to assess among others the linear relationship between altitude and incidence rates and incidence and population density. First of all, this test is valid when both variables of concern are normally distributed. Could the authors please confirm whether they assessed normality in their variable distributions? Otherwise a non-parametric test might be more suited. 

Secondly, R2 is listed along side the estimated pearson’s correlation. This could be me, but I would say reporting pearson’s r is more common. Can the authors confirm what is reported is the R2 and why? Now more importantly, the authors report on the significance of their correlation between COVID-19 incidence and population density.

Yes, both variables, altitude, and COVID-19 cases, had normal distributions (Anderson-Darling test A2Altitude=0.458; pAltitude=0.251; A2COVID-19_cases= 0,633; pCOVID-19_cases =0.092). The values of Pearson’s “r” are now reported instead of R2 in both, the main text, and the figures. 

 Although I have nothing against normalising the result by population density, I doubt relying on merely a significant p-value with such a low R2 provides the right ‘prove’ to do so (this might relate to a high sample size, but as listed in the minor comments, it is a bit unclear to me which test is fitted to which data). I think this is also confirmed by the high variance observed in the correlation between these two variables in Figure S2. Perhaps better to explain rational for normalising incidence by population density in the methods and leave out 3.1.

We appreciate the suggestion of the reviewer. We have included the rationale for normalizing the data of COVID-19 cases by population density in the methods section: 

“The number of COVID-19 cases by location was normalized by population density (inhabitants per square kilometer), in accordance with previous studies that demonstrated a positive correlation between the population density and the number of COVID-19 cases [23-26].” 

Also, the section 3.1 of the results was removed as well as Figure S2.

 I find the authors conclusions more concerning for table 2, where significant p values go alongside with a wide range of R2 values. Explanation is in part covered in the discussion section pp 18, but this is for the countries where no correlation is found. I think this could be done more elaborate, among which how quality of passive surveillance could affect the findings in terms of strength of the correlation.

We thank the reviewer for this comment and suggestion. Firstly, table 2 has been modified to include “Pearson’s r” instead of R squared following a previous suggestion from the reviewer. Secondly, in Table 2 we show how the incidence of COVID-19 is negatively correlated with altitude even when analyzing American countries individually. These findings correspond well with our results at whole-continent level (correlation and Random block design ANOVA). Since the results are consistent, we believe that the most important matter to discuss, is why some countries with large populations living above 1,000 masl do not show significant correlations. We agree with the observation of the reviewer saying that the strength of correlation may be affected by the quality of passive surveillance, however, this is a topic challenging to study in the Americas since health policies are very heterogeneous among countries. Also, the information regarding surveillance policies during the COVID-19 pandemic is not always accessible or trustable in these countries, thus any discussion in this regard may result speculative. 

 The authors are speaking of “the SEIR” model, but in fact, SEIR model structures can involve a multitude of assumptions and parameters encompassing these assumptions. As a result, I have some difficulty assessing the validity of the findings regarding the evaluation of the virus transmission rates.

 Therefore, first of all “an SEIR model structure” would be more appropriate on pp6 line 122.

The correction has been made, now the text reads: “A SEIR model (Susceptible - Exposed - Infectious - Removed) was used to…”

 Also, a listing of the differential equations either in the methods section or the supplementary material would be useful to provide the reader transparency in what model parameters and assumptions were incorporated. E.g. what is assumed regarding the transmission rate of asymptomatic individuals? Non infectious at all? What fraction of the population is assumed to remain asymptomatic?

The equations, parameters, values, and the sources of the values used in our model are now described in the supplementary material S3. Also, the following text has been added in the methods section: 

“We set the values of interaction frequency = 8.1 [28], infectious period = 7.5 days, and incubation period = 6 days [29]. Asymptomatic individuals were considered as non-infectious. Recovered individuals were considered to be immune to reinfection. The size of the population was considered unchanged during the modelled time lapse.”

 Also, it is good practice to list the parameters values used as well as their source. Therefore, I would argue for a table in the supplementary material.

Please see the response to the comment 2.4.2.

 Moreover, the interaction frequency needs explaining. A more commonly used term I think is “contact rate”. The authors refer to beta as their contact rate, but this would more commonly be coined “the effective contact rate” (which can be denoted as a product of contact rate and probability of infection upon contact).

We named the terms as described in (Abadie, Bertolotti, & Arnab, 2020). Indeed, in our manuscript beta is referred as the “contact rate” according to the following equation: 

β=interaction frequency among individuals*probability of transmission of the disease

Where:

β: Contact rate

The description of the parameters is now included in S3.

 More importantly, heterogeneity in transmission of SARS-CoV-2 has been outspoken (e.g. https://wellcomeopenresearch.org/articles/5-67/v3, https://doi.org/10.1101/2020.08.09.20171132 ), which has among others been associated to heterogeneities in age-specific contact patterns. Therefore, estimating an R0 based on an average contact rate, probability of transmission as well as infection period is not a very accurate representation of SARS-CoV-2 dynamics. Importantly, quantifying differences between country-level altitude, by “playing around” with the transmission rate does not come across methodologically sound and could, at minimum, do with a formal fitting procedure.

We calculated R0 values as they were requested by another reviewer. Moreover, as described in the text (pp 6 line 121), the data we analyzed was taken during a period of strict quarantines in Argentina, Bolivia, Colombia, Ecuador, and Peru, when the mobility of people was drastically reduced. So, it is fair to assume that the heterogeneity in contact rate among age groups also decreased. 

Our models were originally adjusted using the least squares method, however, attending the accurate observations of the reviewer, we have adjusted our models again by using the maximum likelihood method. Also, we have calculated the confidence intervals of the estimated transmission rates. Although the estimated values of probability of transmission differ slightly from those we reported in the original version of our manuscript, the trends and conclusions remain unaltered (except for Colombia).

These procedures together with the new values were added/replaced in the corresponding sections of the manuscript, including Fig 3 and Table 3.

 Moreover, in my opinion the differences between transmission rates in the high and lowlands are hardly convincing. A deterministic model is used, and confidence intervals are missing, but my suspicion is that these would highly overlap.

We appreciate the observation of the reviewer; however, we want to emphasize that we set our SEIR models to match the data of cases officially reported by May 23rd for the highland and lowland regions of each country (dotted lines in Figure 4). In our exercise we aimed to replicate the reported data, not to predict future trends. In this context, a deterministic model should work well. Moreover, as the reviewer can see in Table 3, the confidence intervals calculated for the estimated transmission rates do not overlap.

 What I am also not certain about (but I might have missed it), do the authors vary the “frequency of interaction rate” between high and lowlands? As the SEIR model is ‘fitted’ to raw incidence, one would expect different frequency of interaction rates in higher than in lower densely populated areas. In a dynamic transmission model where beta, the effective contact rate, is a product of the frequency of interaction and the probability of infection upon contact, the ‘fitted’ value for the latter will be correlated to the value used for the former. This needs clarification.

As mentioned above, the data we analyzed corresponded to the period between the first reported case for each country (all of them around March 10th) and May 23rd, a period during which strict quarantines were applied in theses countries, therefore, the difference in contact rates between populations with high and low population densities that would occur in regular conditions was drastically reduced.

 In conclusion, I feel rather uncomfortable by the SEIR model used and validity of the findings reported on the transmission rates and R0 estimates between highlands vs lowlands. I doubt these findings should be part of the manuscript.

We are aware that using a deterministic model implies limitations, however, we believe that the differences we found in transmission rates between highlands and lowlands are very clear. This is supported by the small confidence intervals we found for the estimated parameter. So, even if the values we estimated for the transmission rates and R0 are not precise, the epidemiological trends should not be considerably different. We are convinced that this part of our work offers important support to the hypothesis of an attenuated effect of COVID-19 in high regions. 

 Estimate for severity is based on the recovered to case ratio and recovery rates. For clarification, what do the deaths represent in the death to case ratio? The national reported covid deaths? Excess mortality? As I am not entirely sure why there should be such a gap between 1- fraction of of recovered patients (recovered/total cases) and death to case ratio (deaths/total cases).

Deaths represent the total deaths reported for the corresponding altitude group: i.e., above 1,000 and below 1,000 masl. Accordingly, the recovered patients represent the total recoveries reported for the corresponding altitude group. Thus, the gap the reviewer mentions, represents the fraction of active cases, those which are not dead nor recovered either. 

The text in section 2.4 of the manuscript has been modified accordingly: 

“The death-to-case ratio and the percentage of recovered patients ([recovered patients/reported cases] * 100) for each country (except Ecuador) were calculated using the data from the last 10 days evaluated (from May 13th to 23rd) for the populations above and below 1,000 masl in two separate pools. The number of deaths and recoveries used to calculate these parameters are the summatory of the values reported for all the populations above and below 1,000 masl.”

 Regardless, both are very likely sensitive to underreporting (i.e. the higher underreporting, the higher the death-to-case ratio). This is the reason why the authors evaluate the underreporting of cases in high and lowlands. The authors conclude that the non-significant differences observed in death-to-case ratio between high and lowlands could be explained by differences in in undiagnosed cases (76% for highlands vs 73% for lowlands). Estimating confidence intervals based on mean and SD will, I think, show that these estimates overlap, i.e. revealing that this might not necessarily explain the non-observed difference. Even if there was a true difference in underreporting between high and lowlands, why would one expect a difference in underreporting based on altitude? Please clarify and what could be an alternative explanation. In particular why there is a difference in %recovered but not in death to case ratio (but as stated, I don’t fully understand the difference).

The reviewer is right, both indicators are very sensitive to underreporting, specially considering the limitations that these countries had for testing people during the first months of the pandemic.

The underreporting would be directly associated with a higher proportion of asymptomatic, mild, and moderate cases occurring in the highlands as a result of the lower severity of the disease in these regions. Due to the scarcity of tests, during the first months of the pandemic, diagnosis was favoured to people showing clear symptoms or reporting recent contact with infected subjects. Asymptomatic, mild, and a fraction of moderate cases would not be diagnosed. 

Mortality rates may be indicators of the access to ICU facilities, opportunity and quality of clinical treatment, and the severity of the disease in severe and critical patients. While the recovery rate may include the effectiveness of out-of-hospital (pharmacological interventions) treatment (most common strategy to treat COVID-19 in Latin America), as well as the factors related with mortality when concerning severe and critical cases. In this context, no differences in mortality rates but a higher recovery rate in the highlands compared to lowlands, suggest again a lower severity, at least in asymptomatic, mild, and moderate cases, of COVID-19 in the highlands.

 Also methods pp 7 lines 149 – 151: COVID severity: These seem to be based on national estimates (npairs = 5). This while the estimates for the correlation between cases and altitude and case numbers are based on a mixture of national, regional and local level data (see minor comments). Why the difference? To what extend does it make sense to use national level data here, why not, similar to the analyses correlating altitude with incidence, on a more granular level?

We wanted to evaluate this data using a paired design, so we could eliminate inter-country effects. Considering this, it would be very difficult (if not impossible) to pair, objectively, states from the lowlands with states from the highlands. In consequence, we decided to pool the data from the lowland states and the highland states in two separate datasets for each country (5 countries), then we used these pairs to evaluate the differences in recovery and mortality.

As stated in the manuscript, these analyses were made at state level.

 Discussion pp 19: This seems to elaborate quite far on what is covered in the manuscript. Virus transmission capacity under different altitudes I think is what the authors want to elude to, but it now comes across somewhat as a self-citation exercise and covering a topic the authors is probably familiar with. Suggest to cut and/or shorten.

We appreciate the comments of this reviewer. However, we think that, since a limited amount of literature is available on the subject, it is important to report as much as possible our epidemiological findings and make links with likely theoretical and physiological explanations to help in the building of a better understanding of the particular behaviour of COVID-19 in highlands. 

 MINOR COMMENTS

 Methods pp 5 line 105: what do the n=51 represent? Is this a mixture of observations on city or province or state or country or departmental level?

Two changes have been inserted in the text to clarify this point:

 “The number of COVID-19 cases by location (per city/county or per state/province/departamento) was normalized by population density (inhabitants per square kilometer), in accordance with previous studies that showed a positive correlation between the population density and the number of COVID-19 case.” 

 “The correlation between the number of COVID-19 cases per 100-meters-of-altitude interval and the altitude was analyzed using a Pearson correlation analysis (n= 51).”

 Similar for pp 6 lines 109-110, what do the n’s represent here? It seems country-level datapoints, but as the block variable concerns ‘country’ there must be multiple observations from within the countries included. Moreover, these are estimates all across the whole world, not only the American continent?

This analysis was performed at continental level (23 countries). Each data point represents the number of COVID-19 cases at a certain altitude within one country. These altitudes correspond to each city/county or state/province/departamento (according to the availability of data) where COVID-19 cases were reported. This explains the great number of data points. The text has been modified accordingly:

“The dependent variable was the number of COVID-19 cases (at 2nd or 3rd administrative level and not grouped by altitude intervals); the grouping variable was the altitude (> 1,000 masl or < 1,000 masl)”

 And what about the n’s listed in line 111-115 for those countries in the American continent? Please clarify.

These “n” values represent the number of data pairs analyzed in the correlation for each country. The text describing these tests has been moved next to the description of the initial Pearson correlation for clarity. 

 For the non-statistical reader, it would be good to refer to “the block variable” in a more intuitive way.

We thank the reviewer for this suggestion. We have changed the sentence for: “…and the blocks were the countries.” As we do not see a better way to explain the design. If the reviewer has a better suggestion, we will gladly accept it.

 pp 6 line 107-115, terms like “the analysed variable” are not conventional terms to use, rather something along the lines of “dependent variable”.

The text has been changed to: “The dependent variable was the number of COVID-19 cases…”

 “organised by intervals of 100m of altitude”. ‘Categorised’ or ‘grouped by’ might be preferred.

The text has been changed according to the suggestion of the reviewer: “These data were then grouped by intervals of 100 meters of altitude.”

 Pp 6 line 122: The SEIR should be An SEIR. There is not just one SEIR model.

Changes have been made accordingly throughout the text:

Pp 6 line 125 “A deterministic SEIR model (Susceptible - Exposed - Infectious - Removed) was used…”

Pp 13 line 274 “This causes the requirement of a higher infection probability value in our SEIR model to fit…”

 There are certain sections in the result section, that are better suited in the discussion, as these provide interpretation of the results in the context of existing evidence, i.e.:

 Results pp 10 line 196 – 197. This should be moved to the discussion.

This section has been removed.

 Results pp 12 lines 258-261: This should be moved to discussion.

The text has been modified. Now it reads: 

“Taken together, these findings show that a significant decrease in the incidence of COVID-19 starts above 1,000 m of altitude.”

 Also, no need to repeat again that cases were normalised by population density. The same holds for pp 10 lines 203-204. The methods clearly describe what scale was assumed for the correlations, no need to list this again.

This was corrected throughout the text. Also, the following text has been added in the Methods section:

Lines 104-105: “These data (referred as the number COVID-19 cases) were used for all the analyses unless stated otherwise.”

 Methods pp 10 line 207: I think a word is missing in this sentence, but also I don’t fully understand what is meant. I think that beyond 1000 masl, a correlation between altitude and COVID-19 incidence is apparent (and not below) but this is not what it reads.

The reviewer is right, there was a typo in the sentence. The text has been changed accordingly:

“No significant correlation was found for data below 1,000 masl (p=0.568; r= -0.206) (Fig 1c), while a strongly significant correlation between COVID-19 incidence and altitude was obtained for altitudes above 1,000 masl…”

 I also don’t understand what the authors have done when they state “repeated correlation analyses performed at altitudes above 800, 1000 etc.”

The text has been changed to: “In separate correlation analyses considering data from altitudes above 800, 1,000, 1,500, and 2,500 masl, we confirmed this observation.”

 Methods pp 11 lines 219 – 221: Move to methods.

The sentence “The advantage of this type of statistical analysis is that it considers the internal variability of each country in the incidence analysis.” Has been moved to methods.

 Figure 1B legend: add “above 1000m”?

The panels in the figure have been reorganized. Now the legends match correctly with the panels.

 Fig 2: It might be the resolution, but I fail to see the blue circles. Should these be ‘red circles’?

The reviewer is right. “Blue circles” has been changed for “Red circles” in the legend of Figure 2.

 TYPOS

 Introduction pp4 line 76: Appears to have referencing non-consistent with the remainder of the article

The reference has been formatted accordingly.

 Introduction: It’s not very common to include tables in the introduction. Is this table really needed? Or can it be moved to Supplementary material?

Table 1 has been moved to supplementary material (now it is S1) and the text has been changed accordingly.

 Methods pp 8 lines 160-165: Sentence is not correct.

The sentence has been restructured. Now it reads: “Since health policies in most countries in the American continent restricted the access to COVID-19 tests to people showing clear symptoms of infection or with history of contact with infected people…”

 Methods pp 8 line 167: Ref 28 (worldometer) the correct reference?

The reference has been updated using the citation format suggested by the web platform: “Worldometers.info. Coronavirus Death Rate (COVID-19) - Worldometer Dover, Delaware, U.S.A.2020 [19/05/2020]. Available from: https://www.worldometers.info/coronavirus/coronavirus-death-rate/.”

 REFERENCES CITED IN THIS LETTER

Abadie, A., Bertolotti, P., & Arnab, B. D. (2020). Epidemic Modeling and Estimation.

---

## [Editor Report · Decision Letter 2]

3 Mar 2021

Decreased incidence, virus transmission capacity, and severity of COVID-19 at altitude on the American continent

PONE-D-20-23585R2

Dear Dr. Soliz,

We’re pleased to inform you that your manuscript has been judged scientifically suitable for publication and will be formally accepted for publication once it meets all outstanding technical requirements.

Kind regards,

Kristien Verdonck

Academic Editor

PLOS ONE
---

## [Editor Report · Acceptance letter]

15 Mar 2021

PONE-D-20-23585R2 

Decreased incidence, virus transmission capacity, and severity of COVID-19 at altitude on the American continent 

Dear Dr. Soliz:

I'm pleased to inform you that your manuscript has been deemed suitable for publication in PLOS ONE. Congratulations! Your manuscript is now with our production department. 

Kind regards, 

on behalf of

Dr. Kristien Verdonck 

Academic Editor

PLOS ONE